# Prototype Network for Predicting Occluded Picking Position Based on Lychee Phenotypic Features

Yuanhong Li [1,2,3], Jiapeng Liao [1], Jing Wang [1], Yangfan Luo [1,*] and Yubin Lan [1,2,3,*]

1 College of Electronic Engineering (College of Artificial Intelligence), South China Agricultural University, Guangzhou 510642, China; liyuanhong@stu.scau.edu.cn (Y.L.); jiapengliao@stu.scau.edu.cn (J.L.); jingzai@stu.scau.edu.cn (J.W.)
2 Guangdong Laboratory for Lingnan Modern Agriculture, Guangzhou 510642, China
3 National Center for International Collaboration Research on Precision Agricultural Aviation Pesticides Spraying Technology (NPAAC), South China Agricultural University, Guangzhou 510642, China
* Correspondence: fit@stu.scau.edu.cn (Y.L.); ylan@scau.edu.cn (Y.L.); Tel.: +86-13380057253 (Y.L.)

**Abstract:** The automated harvesting of clustered fruits relies on fast and accurate visual perception. However, the obscured stem diameters via leaf occlusion lack any discernible texture patterns. Nevertheless, our human visual system can often judge the position of harvesting points. Inspired by this, the aim of this paper is to address this issue by leveraging the morphology and the distribution of fruit contour gradient directions. Firstly, this paper proposes the calculation of fruit normal vectors using edge computation and gradient direction distribution. The research results demonstrate a significant mathematical relationship between the contour edge gradient and its inclination angle, but the experiments show that the standard error projected onto the Y-axis is smaller, which is evidently more conducive to distinguishing the gradient distribution. Secondly, for the front view of occluded lychee clusters, a fully convolutional, feature prototype-based one-stage instance segmentation network is proposed, named the lychee picking point prediction network (LP³Net). This network can achieve high accuracy and real-time instance segmentation, as well as for occluded and overlapping fruits. Finally, the experimental results show that the LP³Net based on this study, along with lychee phenotypic features, achieves an average location accuracy reaching 82%, significantly improving the precision of harvesting point localization for lychee clusters.

**Keywords:** gradient distribution; lychee; instance segmentation; mask; fault-tolerance





## 1. Introduction

Lychee, a prevalent subtropical fruit predominantly cultivated in southern China, boasts an annual yield exceeding 1 million tons. Guangdong province contributes over 50% of China's lychee cultivation and production [1]. In response to evolving labor paradigms, the agricultural workforce is on a persistent decline, rendering the adoption of harvest automation robots a pivotal approach for agricultural advancement [2]. Furthermore, the mechanization of lychee harvesting holds immense potential to mitigate labor shortages. This statement reflects the depth characteristic of a scientific research paper.

The algorithm of litchi picking point recognition is the key factor affecting the performance of the litchi harvesting robot visual recognition system. In the last three years, the application of artificial intelligence in agriculture has shifted from other fruits to litchi. Some studies used DeepLabV3 to segment and identify litchi fruits or branches [3–5], and some studies used yolo technology to detect and identify litchi [6–9]. For lychee clusters with stem diameters obscured from leaf occlusion, it is challenging to locate them based on texture in RGB images [10–13]. Neural networks struggle to learn positional features for occluded lychee clusters. However, humans can often predict harvesting points based on their own experience. What information is this reliance based on? This paper aims to address this issue via fruit growth morphology and mathematical distribution probabilities.

Current instance segmentation algorithms based solely on deep learning have two disadvantages: (1) they require significant computational power and are difficult to achieve real-time detection, and (2) they struggle to accurately locate stem diameters for occluded lychee clusters [14–16]. Wu et al. [17] devised an approach for extracting 3D contour features from fruits. This approach involves grouping fruit point clouds via the Conditional Euclidean Clustering algorithm and subsequently employing Random Sample Consensus (RANSAC) for spherical segmentation. Li et al. [18] proposed a multi-task perception network for the instance segmentation and detection of calyx and main stem in cherry tomatoes. The network utilized a dual-branch loss function to balance multi-task learning and constructed a Classification and Regression Tree (CART) model. The results showed that the proposed network achieved an F1 score of 95.4% for detecting calyxes, and the average precision, for instance, segmentation of the stem and main stem were 38.7% and 51.9%, respectively. Zhao et al. [19] proposed an adaptive learning method to achieve an output-feedback robust tracking control of the systems with uncertain dynamics, constructing an augmented system using the system state and desired output trajectory. Clearly, the adaptive learning method can effectively address the problem of locating fruit harvesting points, combining parameters such as morphology and gradient direction distribution, allowing the instance segmentation network to more accurately identify and segment occluded and overlapping target images. Liu et al. [20] introduced an emerging Graph Structure Learning (GSL) method, Evolutionary Graph Neural Network (EGNN), designed to enhance the performance of Graph Neural Networks (GNNs). Evidently, EGNN's evolutionary strategy enhances its defense against attacks, which could be beneficial when dealing with lychee image data that may have inherent noise and incompleteness. It also aids in handling the diversity and complexity of lychee images. However, EGNN's evolutionary process may introduce significant computational complexity, especially on large-scale lychee image datasets. This might require additional computational resources and time. Therefore, a careful assessment of its computational resources, data, and performance requirements is necessary before application. Additionally, it should be compared and validated against other conventional methods to determine its actual benefits in lychee image processing. Wang et al. [21] proposed the use of heterogeneous network representation learning to handle data with different types or attributes and map them to a shared low-dimensional representation space. While this approach exhibits advantages in certain domains, it requires careful consideration of its strengths and weaknesses in lychee image object detection and instance segmentation tasks. Firstly, heterogeneous network representation learning is applicable to various types of data, allowing simultaneous processing of lychee images and related text or other data types to integrate information for object detection and segmentation. Secondly, it effectively merges information from different data sources, contributing to better model generalization across different types of lychee datasets. However, heterogeneous network representation learning typically involves handling multiple data types, which may introduce significant computational complexity, especially on large-scale datasets. Furthermore, designing a heterogeneous network suitable for lychee detection requires extensive experimentation and tuning, demanding domain expertise and experience. In summary, most existing stem diameter instance segmentation methods are not suitable for lychee due to the widespread occlusion of harvesting points [22–24]. This paper aims to seek a computational method that can fundamentally enhance the detection of occluded harvesting points.

Instance segmentation algorithms can be broadly classified into two categories: two-stage and one-stage methods [25]. Two-stage algorithms, such as Mask-RCNN [26] and other state-of-the-art (SOTA) methods, follow a similar two-stage structure. Mask-RCNN generates binary masks for each RoI while simultaneously performing tasks related to class classification and box offset regression [27]. SOTA two-stage instance segmentation models heavily rely on feature localization for mask generation. They perform feature pooling or alignment within RoIs and then feed the extracted features into the mask prediction network. Due to the sequential nature of these methods, their speed improvement is limited. On the other hand, one-stage instance segmentation methods, such as FCIS [28],

can execute these steps in parallel. However, extensive post-processing is required after instance localization, making it challenging to achieve real-time segmentation. The one-stage instance segmentation algorithm YOLACT introduces mask coefficients parallel to the RetinaNet classification and regression branches. It utilizes channel-wise weighting coefficients to synthesize instance masks and applies a nonlinear transformation to the predicted coefficients [29]. Compared to the two-stage methods, YOLACT eliminates the process of generating local feature maps using RoI Align, resulting in a more streamlined network and real-time speed.

The architecture presented in this paper draws inspiration from Prototype Generation, with the goal of creating an encoder that predicts a set of k prototype masks covering the entire image [30,31]. The input image is mapped into a high-dimensional feature space, where each class's prototype vector is represented by the mean vector of its support set samples. Subsequently, the Euclidean distance between the query sample and the prototype vectors of each class serves as the foundation for determining class attribution and constructing the loss function. Kim et al. [32] proposed a chest radiography framework called XProtoNet for global and local interpretable diagnosis. XProtoNet learns representative patterns for each disease from X-ray images and diagnoses given X-ray images based on these prototypes. The difference between XProtoNet and ProtoPNet [33] is that it can learn characteristics within a dynamic region. The reason for adopting XProtoNet in this study is its robustness in the occluded lychee harvesting region, as the network demonstrates strong performance with prototype features. Zhang et al. [34] proposed an improved grape cluster image segmentation algorithm using adaptive morphology. It defines the edge distance based on the minimum distance between edge points in the minimum domain and disconnected components. The algorithm utilizes an improved region classification algorithm with multiple principal components. The average precision of grape stem segmentation and extraction improved by 9.89% and 2.17%, respectively. However, this method lacks robustness for different stem diameters and does not address the localization of occluded or overlapping harvesting points.

## 2. Materials and Methods

### 2.1. Image Acquisition

The dataset used in this paper consists of a total of 5800 lychee images with a size of $1440 \times 1080$ and 400 with a size of $1920 \times 1080$ images with RGB-D information. These data were collected from lychee orchards in Conghua, Guangdong, China, and included only two varieties, namely Heiye and Feizixiao. The images were captured at a distance of approximately 350–450 mm from the lychee, and the camera lens plane was aligned as closely as possible to the frontal view of the lychee fruit cluster's center, without any top or bottom views. During data labeling, the fruit contours were initially marked, followed by the addition of two-dimensional coordinates (x, y) for the occluded picking points. We divided the lychee picking point location into two scenes based on visual observation [35–40]. If the occlusion area of leaves or branches exceeded 30%, the sample was considered occluded and labeled as type A. Unobstructed samples were labeled as type B. Typically, the picking points are distributed along the mid-line of the geometric center [41,42]. However, since a single lychee fruit weighs about 21.4–31.8 g, the weight can cause the fruit to easily lean to one side due to gravity [43–45].

### 2.2. Coarse and Edge Computation of Lychee Morphology

Lychee fruit cluster images exhibit complex edges and holes, and instance segmentation can separate the fruit entities. Existing methods require significant computational resources to scan the image and use equivalent sequences to record labels of connected components in adjacent rows, such as contour-based and quadtree-based methods [34]. This paper proposes a minimum domain computation method that can handle situations with different labels and unordered labels without the need for equivalent sequence processing. It only requires a single scan of the image to obtain disconnected components with different labels [46]. The specific steps are as follows: (1) Initially, the image undergoes row-by-row scanning. Within

each row, any nonzero element is gathered to construct a 1-dimensional array. The positions of these nonzero pixel values are documented as labels. (2) For every row, except the first one, an evaluation is made to determine whether the current run is linked to any of the $n$ (where $n$ is set to 5 in this study) neighboring runs in the row just preceding it. When no connection is found, a new label is assigned to the identified run. In this case, the labels of runs in the previous row remain unaffected. If only one connected run from the previous row is identified, the current run is labeled with the same label as the connected run. (3) Ultimately, by executing the described steps, it is possible to assign labels to all edge pixels in the lychee fruit cluster image that are not interconnected.

After obtaining the distribution of the image edges, the edge distance is computed using the minimum domain centered on each edge point. The minimum domain includes the edge point itself, the connected components containing the edge point, and the disconnected components from the connected components. The specific process is illustrated in Figure 1. Given a segmented image of size W × H, *Fn* sequential points on the fruit edge are obtained. Initialize parameters $i = 0$ and $j = 5$; $Max_{mn} = \max(m, n)$, where $m$ and $n$ are the dimensions of the domain. In the domain $M_{ij}$, disconnected components are detected. The center point of the domain serves as the radius for calculation, with $j$ ranging from 5 to $Max_{mn}$, where $Max_{mn}$ is the larger of the two dimensions. It is checked whether there are disconnected regions in the domain. The Euclidean distance $Dr$ between point $i$ in the domain and the unconnected point $j$ within the domain is calculated. The $Dr$ array is traversed to find the minimum value $D_{min}$, which is then assigned as the edge distance for the pixels in that domain. These steps are repeated until the $D_{min}$ for all *Fn* points in the image is obtained.

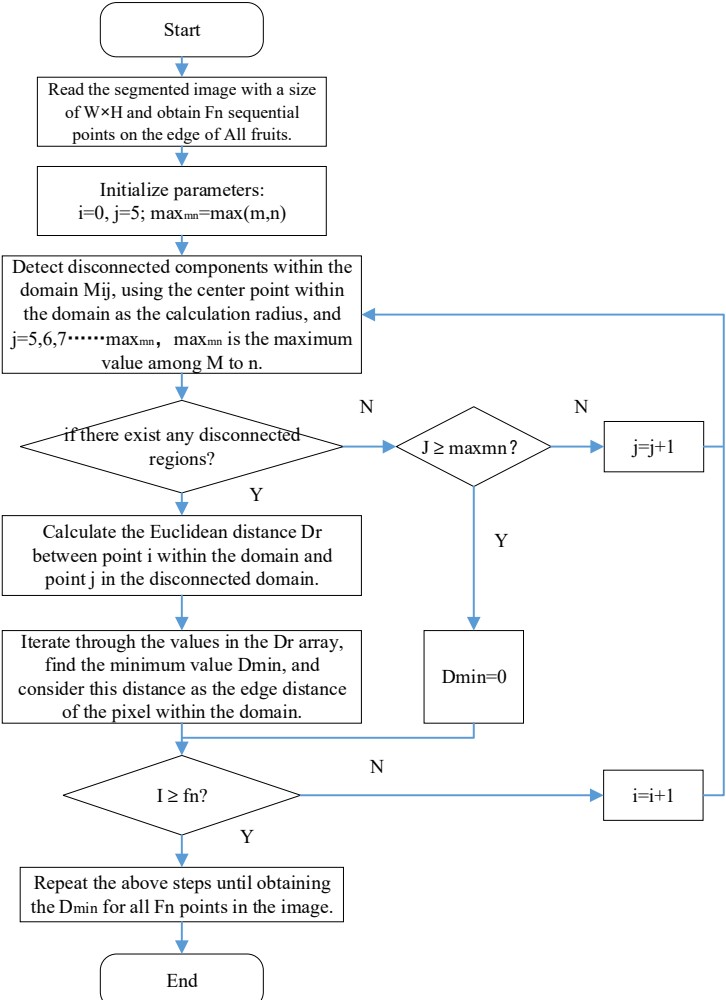

**Figure 1.** Calculation of edge distance within the domain.

### 2.3. LP³Net Network Design

Lychee fruit clusters are different from grapes, cherry tomatoes, and other fruits. This is because a single lychee fruit weighs around 21.4–31.8 g, and the number of fruits in a lychee cluster can range from 3 to 15. The weight of the fruits makes the lychee cluster prone to sagging under the influence of gravity and susceptible to leaf occlusion. Moreover, the actual picking point P′ forms an inclined angle between the predicted picking point P on the midline of the cluster and the main stem. Therefore, instance segmentation algorithms for lychee require high accuracy to effectively handle occlusion and overlapping masks. You Only Look At CoefficienTs (YOLACT) primarily addresses the issue of slowed ROI Pool/Align and segmentation in the two-stage Mask-RCNN [29,47]. Inspired by YOLACT, this paper proposes LP³Net, an improved instance segmentation algorithm based on YOLACT. First, the backbone network utilizes ResNet101 as its main network for extracting feature representations from input images. On top of the backbone network, the Feature Pyramid Network (FPN) is employed to generate a multi-scale feature pyramid. As shown in Figure 2, feature P5 is obtained from the C5 layer via a convolutional layer, and then bilinear interpolation is used to double the size of the feature map. The feature map C4 is added to obtain P4. Moreover, P3 is passed to XProtoNet, and P3 to P7 is simultaneously sent to the prediction head. Each prototype corresponds to a mask coefficient according to references [33,48]. Each anchor returns (4 + n + k) coefficients, which include 4 coordinate coefficients and the corresponding category. Next, in this paper, XProtoNet is used to generate instance-level feature representations. XProtoNet consists of a series of 2D convolutional layers that are used to generate feature vectors for each instance. Following ProtoNet, LP³Net utilizes a series of prediction heads to predict the category and mask of the targets. Each prediction head consists of convolutional layers and fully connected layers to extract features and generate corresponding prediction results. Additionally, LP³Net incorporates a Detection Head for the localization of individual lychee fruits, allowing for the calculation of positional tolerance distance using a few computational parameters based on the detected bounding boxes. Apart from predicting the category and bounding boxes, LP³Net also includes a segmentation Mask Head for generating pixel-level segmentation masks of the targets. This head utilizes convolutional and upsampling operations to generate dense segmentation masks for the targets. During the training phase, the boundary refers to the ground truth bounding box, while during the evaluation phase, it refers to the predicted bounding box. The threshold value of 0.5 is used to perform image binarization on the generated mask. Finally, LP³Net undergoes a position processing step for target point tolerance localization, which integrates the calculation of minimum domain edge and tolerance distance for individual lychee fruits.

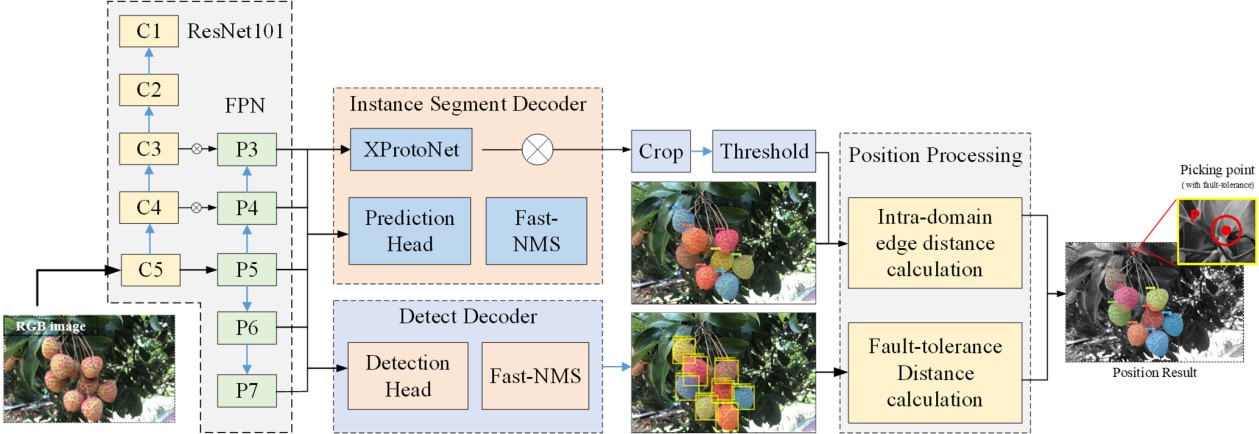

**Figure 2.** LP³Net network architecture (P is the picking point).

### 2.4. XProtoNet

The difference between XProtoNet and ProtoPNet lies in their ability to learn features within a dynamic region [49]. In ProtoPNet, the prototypes are contrasted with feature patches of a consistent size extracted from the feature map. ProtoNet consists of a conventional convolutional neural network f, followed by a prototype layer and fully connected layers. Assuming the CNN model output is H × W × D, the number of output channels, D, in this paper, can be 128, 256, or 512. After computing the scores for all prototypes, a fully connected layer is used to map the prototype scores and the final decision scores [50,51]. As shown in Figure 3, ProtoPNet compares feature patches from all spatial locations of the feature map with the prototypes and outputs the maximum value as the similarity score.

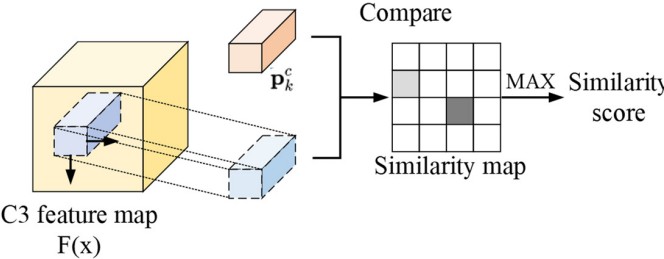

**Figure 3.** ProtoNet obtains similarity to the feature map (Different grayscale regions represent the different prominent features of that area).

The distinctive feature of the LP$^3$Net network is the integration of XProtoNet and position processing. XProtoNet takes into account two independent aspects of the input image: the patterns within the P3 layer shown in Figure 4 and the region of interest focused on the fruit. Assuming the feature map can be represented as $F(x) \in \mathbb{R}^{H \times W \times C}$, where $H$ represents the height, $W$ represents the width, and $C$ represents the number of channels. The pixel region where the fruit appears is represented as each prototype $P_k^c$ to predict the potential feature map $M_{P_k^c}(x) \in \mathbb{R}^{H \times W}$. The feature map represents the most likely locations for individual lychee fruits to appear. After undergoing a $1 \times 1$ convolution, here we compare the feature vectors $f_{P_k^c}(x)$ and prototype $P_k^c$:

$$f_{P_k^c}(x) = \sum_u M_{P_{k,u}^c}(x) F_u(x) \tag{1}$$

where $u \in [0, H \times W]$ denotes the spatial location of $M_{P_k^c}(x)$ and $F(x)$. For ProtoNet in this paper, XProtoNet is used to concentrate the feature maps, and after training the feature extractor in LP$^3$Net, the prototype $P_k^c$ is replaced with the most similar feature vector $f_{P_k^c}$ in comparison. In the dataset used in this paper, the lychee fruit features in the feature map may not be concentrated in a specific region [52,53]. Therefore, if we compare the features with fixed patches like in ProtoNet, it would limit the accuracy of instance segmentation. XProtoNet effectively addresses this issue by considering patch features as part of the model's predictions without restricting comparisons to a fixed region.

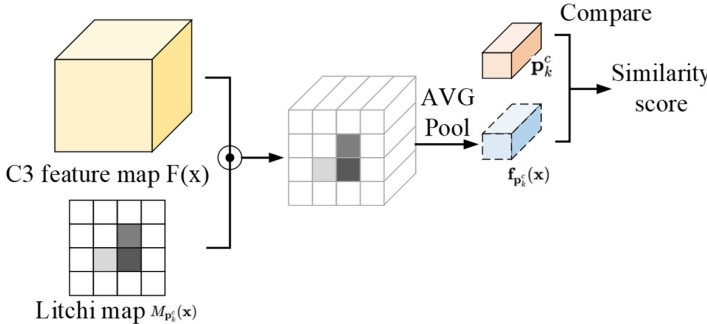

**Figure 4.** XProtoNet obtains similarity to the feature map (Different grayscale regions represent the different prominent features of that area).

### 2.5. LP³Net Loss Function

The loss function of LP³Net in this paper consists of two parts: instance segmentation loss $L_{seg}$ and object detection loss $L_{det}$ [54,55]. During the training phase of the XProtoNet network, $L_{seg}$ is mainly composed of classification loss $L_{s\_cls}$, box regression loss $L_{s\_box}$, and mask loss $L_{s\_mask}$. Specifically, $L_{s\_cls}$ can be defined as follows:

$$L_{s\_cls} = -\sum_i (1 - p_i^c)^\gamma y_i^c \log(p_i^c) - \sum_i (p_i^c)^\gamma (1 - y_i^c) \log(1 - p_i^c) \tag{2}$$

where $p_i^c = P(y^c|x_i)$, $x_i$ represents the score of the *i*-th prediction indicating the presence of an object, and $\gamma$ is an adjustable weight parameter. The mask loss adopts pixel-wise binary cross-entropy loss $L_{s\_mask} = f_{BEC}(x, y)$, which can be defined as follows:

$$L_{s\_mask} = -\frac{1}{N} \sum_{i=1}^{N} y_i \cdot \log(p_i) + (1 - y_i) \cdot \log(1 - p_i) \tag{3}$$

where $y$ is a binary label (0 or 1), and $p_i$ represents the probability of belonging to the y label. In the case of y being 1, if the predicted value $p(y)$ approaches 1, the function value approaches 0. Conversely, if the predicted value $p(y)$ approaches 0, the loss function value will be very large. In summary, $L_{seg}$ can be expressed as follows:

$$L_{seg} = \alpha_1 L_{s\_cls} + \alpha_2 L_{s\_box} + \alpha_3 L_{s\_mask} \tag{4}$$

where $\alpha_1$, $\alpha_2$, and $\alpha_3$ represent the optimization weights for the classification loss, box regression loss, and mask loss, respectively. In this paper, the values of $\alpha_1$, $\alpha_2$, and $\alpha_3$ are set to 1, 1.3, and 3.5, respectively [56,57]. Additionally, the loss $L_{det}$ can be expressed as

$$L_{det} = \beta_1 L_{d\_cls} + \beta_2 L_{d\_box} \tag{5}$$

where $\beta_1$ and $\beta_2$ are optimization weight coefficients for the detection box class and box regression losses, respectively. $L_{d\_cls}$ is the softmax loss for multi-class confidence, and $L_{d\_box}$ uses the smooth L1 loss.

### 2.6. Harvest Target Error Radius Calculation

In order to improve the accuracy of the model predictions, an analysis of the target error radius is conducted based on the cutting position of the end effector. For lychee cluster harvesting, this paper proposes a target localization mechanism based on error analysis. As shown in Figure 5, assuming the angle between the line connecting the predicted point P and the ground truth point P′ and the horizontal line (X-axis) is $\beta$, the cutting angle of the end effector is also set to $\beta$. After the visual system determines the picking point, the end effector checks whether it can cut the lychee bunch stem at point P using force-sensing feedback. If cutting at point P is not successful, the end effector will move along a straight line with distance $D_{end}$ at an angle of $\beta$ for a second cutting attempt to improve the harvesting success rate. Similarly, the picking range of the target point is designed with tolerance at both points P and P′. The target point P is allowed to cut within the effective tolerance radius $R_2$. For occluded picking points, the target point P will be repositioned to P′ for picking by calculating $D_{end}$. Two tolerance distances, $R_1$ and $R_2$, are set in this paper, where the radius of the target circle is dynamically determined by the end effector. Assuming the lateral distance of the end effector is Le, then $R_1 = Le/2$, $R_2 = Le/4$. If the predicted point falls within the range of the radius $R_1$, it is considered a preferred picking (PP); otherwise, if it falls within the circular area with radius $R_2$, it is considered an alternative picking (AP).

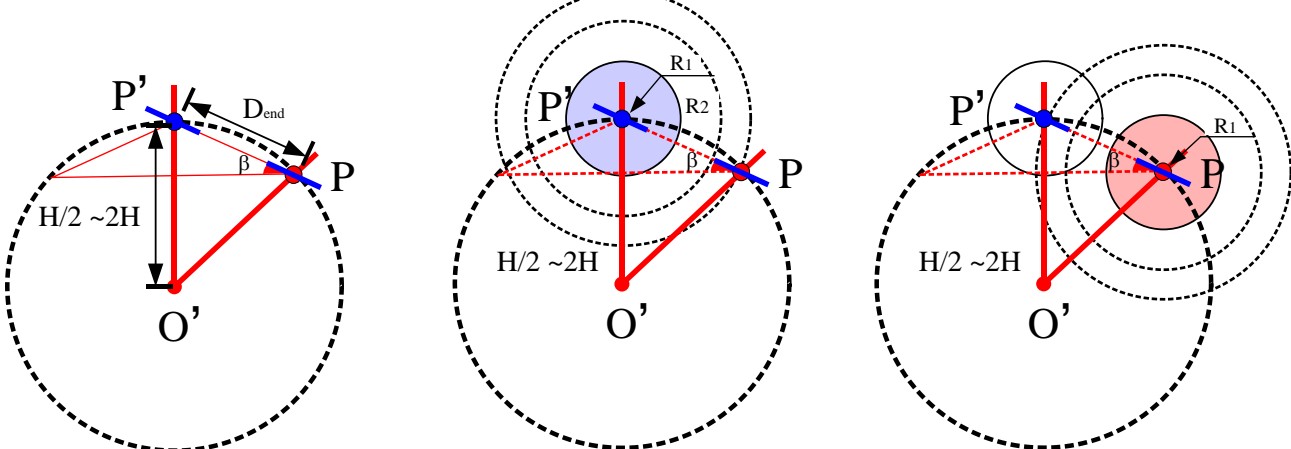

**Figure 5.** Calculation of harvest target error radius.

### 2.7. Gradient Vector Calculation

After instance segmentation via LP$^3$Net, this paper first performs HSV color space processing. Then, using OpenCV with the version number is 3.4.1, the dataset with a size of 1440 × 1080 is proportionally resized to a binary image of size 288 × 216 to reduce computational complexity. As shown in Figure 6, this paper first separates the contour of each individual lychee using the minimum intra-domain and edge calculation. Then, the method for image binarization can utilize the Otsu algorithm techniques, in the following format:

$$T = \underset{T}{\mathrm{argmin}}\left\{\omega_0(T)\sigma_0^2(T) + \omega_1(T)\sigma_1^2(T)\right\} \tag{6}$$

where $T$ is the threshold, $\omega_0(T)$ and $\omega_1(T)$ represent the proportions of two classes of pixels, and $\sigma_0^2(T)$ and $\sigma_1^2(T)$ are the variances of the two classes of pixels. The method for image edge computation can employ the Sobel operator or other edge detection operators, in the following format:

$$G_x = \begin{bmatrix} -1 & 0 & 1 \\ -2 & 0 & 2 \\ -1 & 0 & 1 \end{bmatrix} * \mathrm{I}, \; G_y = \begin{bmatrix} -1 & -2 & -1 \\ 0 & 0 & 0 \\ 1 & 2 & 1 \end{bmatrix} * \mathrm{I} \tag{7}$$

in which I represents the image matrix, $G_x$ and $G_y$ are the gradient matrices in the horizontal and vertical directions, and $*$ denotes the convolution operation. At last, it undergoes discretization and smoothing. Subsequently, the closed contour of the lychee fruit is used to compute the gradient direction distribution. The calculation method for the gradient vector distribution in this paper is as follows: Let the points on the contour line be labeled as $C_i$, where $i$ = 0, 1, 2, ..., g. To improve computational efficiency, we sample the data with a step size of $\Delta K$, resulting in a sampled labeled point dataset $C_{i-k}$, where $i$ = 0, 1, 2, ..., g'. Let $P_i$ be the feature resolution in the length or width direction, and $L$ be the picking tolerance radius of the robotic arm or end effector [58,59]. The value of $g'$ is obtained by dividing $g$ by 10. Then, the calculation can be expressed as follows:

$$1/P_i = \Delta K / L \tag{8}$$

where $P_i$ and $L$ are measured in millimeters, and the step size $\Delta K$ is measured in pixels. Therefore, the formula for computing the gradient direction of the sampled points $C_{i-k}$ on the contour can be expressed as follows (along the X-axis):

$$G_i = [g_x, g_y] = \left[\frac{\partial f(x,y)}{x}, \frac{\partial f(x,y)}{y}\right] \tag{9}$$

$$a(x,y) = \arctan(g_x / g_y) \tag{10}$$

where $G_i$ represents the gradient vector of $C_{i-k}$ along the gradient direction $a(x, y)$. First, contour detection is performed by traversing a range of 9 pixels [60]. The method for gradient direction distribution can involve using histograms or other statistical approaches, in the following format:

$$H(\theta) = \sum_{i,j} \delta\big(\theta - \arctan(G_y(i,j) / G_x(i,j))\big) \tag{11}$$

where $\theta$ represents the gradient direction, $\delta$ is the Dirac function, and $H(\theta)$ represents the histogram of the gradient direction distribution. Then, sampling is conducted from the contour to obtain the gradient of all points, which are then used for statistical analysis. As shown in Figure 6g,h, the fruit contour is separated into two parts from the geometric center for pixel position gradient traversal. The dependent variables for data output are the pixel distributions along the X-axis and Y-axis, respectively. Finally, each contour is labeled with NV according to Section 2.7.

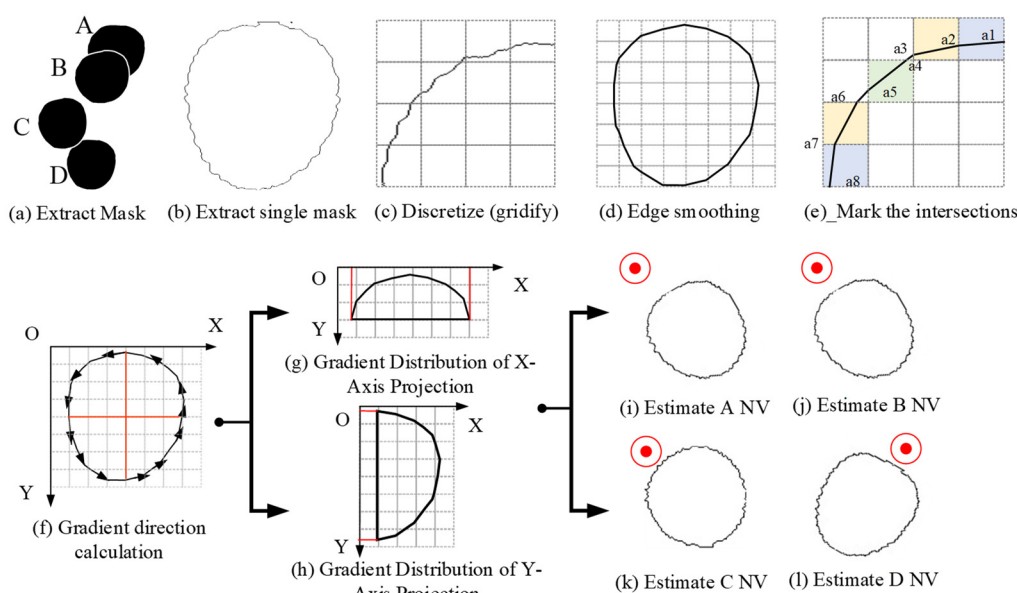

**Figure 6.** Gradient vector calculation A–D represent different identification numbers for four individual lychee fruits, the red portion represents the picking point for individual fruits, while the various colors in (**e**) represent different features).

### 2.8. Statistical Analysis

As shown in Figure 7a, we assume that the origin of XOY coordinate is upper left corner of image. According to left upper corner coordinate ($L_1$, $T_1$) of Bboxs (Bbox) of object detection, and right lower corner coordinate ($R_1$, $B_1$), it can obtain the height H = $B_1$-$T_1$ of the lychee bunches. We marked the assuming picking point as P. Based on empirical data, we can determine the distance between point P and the upper surface of the lychee bunch's bounding box as ranging from H/2 to H. Subsequently, we further estimate Line 1 and Line 3 by both increasing and decreasing this half of this distance by a factor of 1. The Y direction refers to the reserved length of fruit bunches along the main stem, and its fault-tolerant positioning range in this direction is relatively high. It is assumed that the estimated value in the X direction of the picking point P follows a normal distribution. We use Shapiro–Wilke's W test method to verify, and the specific steps are as follows [26,61–64]: (1) With statistical assumption factor $H_0$, the X distance values of the picking points are all from normal distribution. (2) According to the estimated NV value $X_i$ of each bunch of lychee, rearrange $X_1$, $X_2$, $X_3$,..., $X_i$ from large to small. (3) According to

the Shapiro–Wilk coefficient table, find out the Shapiro–Wilk coefficient $\alpha_{in}$ corresponding to the sample size. (4) Calculate the value of the statistic W. First, assume that the sample values are $x_1, x_2, \ldots, x_n$, where n is the sample size and $n \geq 3$. Sort the sample values in ascending order to obtain $x_{(1)}, x_{(2)}, \ldots, x_{(n)}$, where $x_{(1)}$ is the minimum value and $x_{(n)}$ is the maximum value. Then, calculate the sample mean $\overline{x}$ and sample variance $s^2$, and their formulas are

$$s^2 = \frac{1}{n}\sum_{i=1}^{n}(x_i - \hat{x})^2 \tag{12}$$

(a) Coarse localization of Bboxes.  (b) Estimation of height  (c) Estimation of height  (d) All predicted points.

(1) Right distribution  (2) Left distribution  (3) Normality distribution  (e) Output result

**Figure 7.** Gradient vector distribution calculation (Red dots represent the optimal picking points on this horizontal line, while blue indicates alternative picking points).

Next, calculate a set of constant coefficients $a_1, a_2, \ldots, a_n$, and the formula is

$$a_i = \frac{m^T V^{-1}}{\sqrt{m^T V^{-1} V^{-1} m}} e_i \tag{13}$$

where $m = (m_1, m_2, \ldots, m_n)^T$ is the expected order statistic of the normal distribution:

$$V_{ij} = \int_{-\infty}^{\infty}\left[\Phi^{-1}(u)\right]^i\left[\Phi^{-1}(u)\right]^j du \tag{14}$$

where $\Phi^{-1}(u)$ is the inverse cumulative distributed function of the standard normal distribution, and $e_i$ is a $n \times 1$ unit vector, with a first element that is 1, and the rest are 0.

$$W = \frac{\left(\sum_i \alpha_{in}(X_{n+i-1} - X_n)\right)^2}{\sum_{i=1}^{n}\left(X_{(i)} - \overline{X}\right)^2} \tag{15}$$

The numerator $\sum_i$ is $\sum_{i=1}^{\frac{n}{2}}$ when n is even and $\sum_{i=1}^{\frac{n+1}{2}}$ when n is odd. X is the average estimated value of the target NV in the X direction. (5) Select the test level β factor (β = 0.10, 0.05, or 0.01), and obtain the corresponding W(n, β) value according to the number of samples n and the test level factor β difference W distribution table. (6) When w ≤ w(n, β), the overall sample is not normally distributed. If w > w(n, β), the assumed $H_0$ follow a normal distribution [65,66]. Finally, we construct the Shapiro–Wilk distribution learning model by projecting each NV on a line parallel to the X-axis by taking the projected size of the NV ProLi (Yi) and the known picking points. We predict the coordinate position

of Px via supervised learning and then judge whether the picking stem diameter 1 falls within the target circle with $R_2$ as the radius, so there are two expected output values of supervised learning: (1) assuming that the target circle with radius $R_1$ is the PP point, (2) a ring of radius greater than $R_1$ and less than $R_2$ is AP point.

## 3. Results and Discussion

### 3.1. Evaluation of Multiple Models

This article uses average precision (mAP) as the evaluation metric for object detection and instance segmentation, assuming that P is denoted as the actual number of samples among target prediction; this is called precision [67,68]. R is the recall rate, where $P = \frac{TP}{(TP+FP)}$ and $R = \frac{TP}{(TP+FN)}$. The mAP can be calculated via equation $\frac{\sum AP}{N_{classes}}$. Among them, TP represents the number of samples where the predicted category of the model matches the annotated category; FP represents the number of samples where the predicted category of the model does not match the annotated category; and FN represents the number of samples where the model predicts a background class, but the annotated category is another class.

We compare many algorithms for mAP and speed on our dataset and evaluate the detection via single and cluster lychee, respectively [69,70]. We adjusted the dimensions of all images to $288 \times 216$ pixels, considering that their original size was $1440 \times 1080$ pixels. Our hardware devices include an INTEL I7 CPU and NVIDIA GeForce GTX 3060 Ti GPU. For this purpose, we found four embedded development boards that can be used in the picking equipment for testing, namely NVIDIA Jetson Orin, NVIDIA Jetson nano, Orange Pi5P, and Raspberry Pi4B, and gave the test results. Among them, Nvidia devices use GPUs for acceleration, Orange Pi5P because the GPU is not supported by CUDA, so GPU acceleration is not used, and the Raspberry Pi 4b performance is too slow to execute normally. The experiment used Intel RealSense D435i for real-time detection, and the average data were taken within 5 min.

We use the INTEL RealSense d435 to obtain RGB-D images, which are developed by Intel Corporation and integrated with two infrared sensors and an inertial measurement unit (IMU). The CUDA version is 11.0, and the CUDNN version is 7.4. The operating system is Linux with Ubuntu18.04 LTS. All model training is divided into two steps. The first step is to freeze the training, that is, only train the backbone part. The learning rate is set to 0.01–0.001, the number of iterations is 50, and the number of samples for each iteration is 4. The second step of training is the entire detection network, and the initial learning rate is set to 0.001. We trained multiple models using the same dataset and initialization parameters and evaluated their performance using 1000 epochs. In addition, during the training and evaluation process of the independent branch models, we froze the other branch to eliminate interference from multiple branches. As seen in Table 1, we differentiate lychee based on the number of individual fruits. First, in the case of detecting targets with less than five lychee fruits, SSD achieved the best precision with 96.5%, followed by LP$^3$Net with 95.5%. Although LP$^3$Net may not have the same level of accuracy as SSD, it achieves a high recall rate of 94.9%. At an IoU of 65, LP$^3$Net, benefiting from XProtoNet and the positioning process, achieves a mean average precision (mAP) of 80.3%. The detection head of LP$^3$Net can simultaneously predict the category score, bounding box regression parameters, and mask coefficient. In terms of the FPS comparison effect, LP$^3$Net still reached the highest 19.4 fps. Although the mAP of YOLACT is the highest when the IoU is 50, when the IoU value is 65 or 80, LP$^3$Net has a significant effect on improving accuracy. In addition, when the number of lychee fruits is between 5 and 10, the accuracy and recall rate of the eight algorithms for object detection are generally lower due to the presence of mutual occlusion. However, LP$^3$Net still achieves an accuracy of 92.3% and a recall rate of 91.9%. When the IoU value is 65, LP$^3$Net still has a significant effect on improving the accuracy and 18.3 fps in instance segmentation. In summary, the proposed LP$^3$Net in this paper demonstrates good performance in lychee cluster object detection and instance segmentation.

**Table 1.** Comparison of multiple models.

| NO. of Lychee | Network | Object Detection | | | Instance Segmentation | | | FPS |
|---|---|---|---|---|---|---|---|---|
| | | Precision (%) | Recall (%) | F1_Score (%) | mAP | mAP$_{65}$ | mAP$_{80}$ | |
| 0–5 | YOLOV3 | 92.5 | 94.2 | 93.3 | - | - | - | 18.2 |
| | YOLOV5m | 93.1 | 92.4 | 92.7 | - | - | - | 13.4 |
| | EfficientDet | 92.7 | 90.3 | 91.5 | - | - | - | 16.7 |
| | SSD | **96.5** | 90.6 | 93.5 | - | - | - | 10.3 |
| | FasterRCNN | 91.6 | **96.4** | 93.9 | - | - | - | 9.6 |
| | YOLACT | - | - | - | 54.4 | 70.3 | 23.9 | 9.8 |
| | MaskRCNN | - | - | - | 34.6 | 66.2 | 29.7 | 18.2 |
| | LP$^3$Net | 95.5 | 94.4 | **94.9** | 51.2 | **80.3** | **30.3** | **19.4** |
| 5–10 | YOLOV3 | 89.6 | 87.5 | 88.5 | - | - | - | 16.3 |
| | YOLOV5m | 88.6 | 81.4 | 84.8 | - | - | - | 16.6 |
| | EfficientDet | 92.4 | **93.5** | 92.9 | - | - | - | 9.4 |
| | SSD | 94.6 | 92.1 | 93.3 | - | - | - | 8.2 |
| | FasterRCNN | **95.4** | 88.5 | 91.8 | - | - | - | 10.5 |
| | YOLACT | - | - | - | 32.8 | 65.3 | **18.6** | 14.9 |
| | MaskRCNN | - | - | - | 34.6 | 66.2 | 12.5 | 17.2 |
| | LP$^3$Net | 92.3 | 91.6 | **91.9** | **46.5** | **72.1** | 17.2 | **18.3** |

### 3.2. Calculate the Centroid of Contour

In this paper, we counted the number of masks and calculated the MSE of A-type picking. Here, we conducted verification with the number masks of 600, 800, and 1200. Finally, the effective numbers of segmentations for the mask are 562, 720, and 1093. The values of $L_{mask-coefficient}$ are 1.34, 0.91, and 0.82, respectively. When the pixel value is extracted as a unit, the value of $|AB|$ can be calculated using algorithm 1 in this paper. As shown in Figure 8, when the angle of single fruit of lychee is $\mu_1 < 15°$, the mode of WHR is 1.0–1.1; When the $15° < \mu_1 < 30°$ and $30° < \mu_1 < 45°$, we all take the value of WHR as 0.9–1.0; When the $\mu_1 > 45°$, we take the value of WHR as 0.8–0.9. The error between pixels predicted via the $P_x$ point and the distance under the world coordinate was 3.64 cm. The difference between pixels predicted via $P_y$ point and distance under world coordinate was 2.15 cm.

### 3.3. Gradient Calculation and Regression Analysis

To validate the effectiveness of the algorithm, this study first performed gradient analysis on the complete contours of two lychee fruits with stalks. The analysis involved calculating $\arctan(g_x/g_y)$ to analyze the data. Since the contours of fruits with stalks are more distinct than single fruits, it is easier to establish corresponding relationships. In this analysis, the contours of lychee fruits with stalks were traversed at the pixel level, and gradients were calculated using a traversal unit consisting of a 9-pixel grid.

As depicted in Figure 9, the turning points are denoted as $TP_x$, where x represents the number of turning points. Through comparison, as shown in Figure 10, $TP_1$ and $TP_3$ correspond to the vertex positions of the lychee stalk contours, while $TP_2$ and $TP_4$ represent the adhesive edges between the two lychee fruits that have not undergone contour separation via the minimum domain-based edge calculation. To predict the relationship between the inclination angle of the picking point and the contour of a single lychee fruit (or a cluster of fruits), this study conducted a multivariate linear regression analysis using a dataset of 360 lychee fruits. The dataset included the rotation angles of individual fruit contours and their corresponding gradient value distributions. Initially, 129 fruit contour data were selected, and the contours were projected along the X-axis with the morphological center as the midpoint. Nine points were sampled along the contour edge in sequential order. As shown in Table 2, the correlation coefficient between the fruit contour rotation angle and the gradient value distribution was 0.964, with a standard error of 10.298.

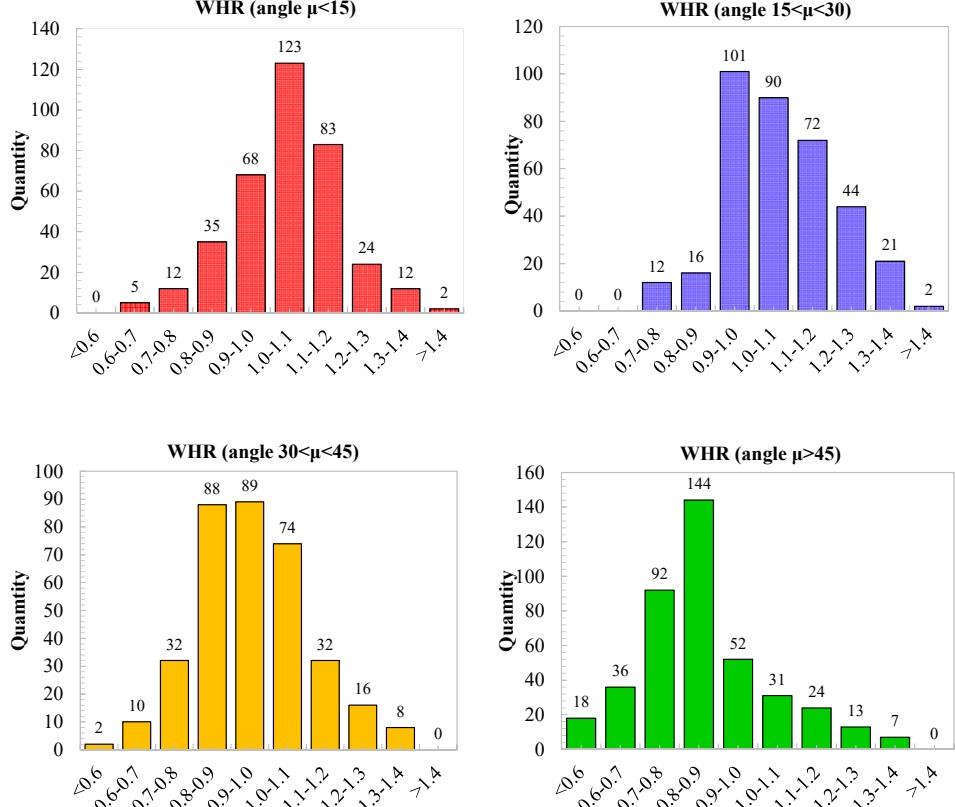

**Figure 8.** Mask-based WHR statistics.

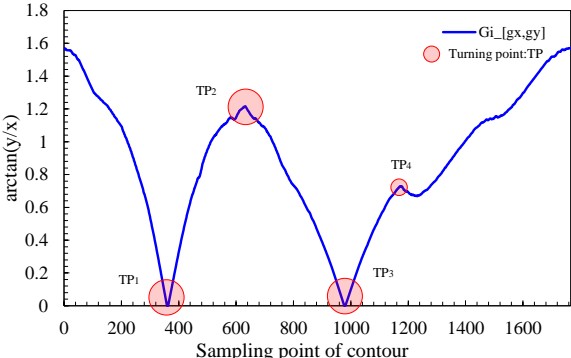

**Figure 9.** Gradient distribution by order.

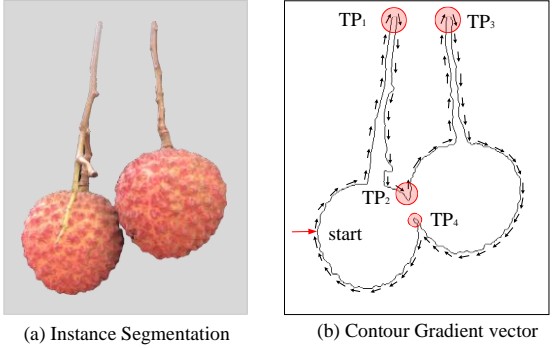

(a) Instance Segmentation          (b) Contour Gradient vector

**Figure 10.** Inflection point contour (The arrows in the diagram represent the starting points of the program's traversal of the contours).

**Table 2.** Regression statistical results.

| Regression Analysis | |
|---|---|
| Multiple R | 0.964078965 |
| R Square | 0.929448252 |
| Adjusted R Square | 0.924112405 |
| Standard Error | 10.29819955 |
| Observations | 129 |

Table 3 is the regression parameter table. From the standard, we can observe that the intercept is 58.80. The smaller the standard deviation, the higher the precision of the parameter. Additionally, most of the parameters in Table 3 have a *p* value less than 0.05, indicating that the model is significant or has a confidence level of 95% at $\alpha = 0.05$. Among them, the *p* values of X variables 1, 2, 3, 4, 8, and 9 are all less than 0.01, indicating a stable mathematical relationship between the rotation angle of the individual lychee contour and its edge gradient distribution. Lastly, the regression coefficients with a 95% confidence interval are provided, where the upper and lower limits of the intercept's variation at $\alpha = 0.05$ are 46.54 and 71.06, respectively.

**Table 3.** Regression parameters of X-axis.

| | Coefficients | Standard Error | t Stat | *p*-Value | Lower 95% | Upper 95% | Upper Limit 95.0% | Lower Limit 95.0% |
|---|---|---|---|---|---|---|---|---|
| Intercept | 58.80 | 6.19 | 9.50 | 0.00 | 46.54 | 71.06 | 46.54 | 71.06 |
| X Variable 1 | 298.76 | 37.14 | 8.04 | 0.00 | 225.22 | 372.30 | 225.22 | 372.30 |
| X Variable 2 | −196.57 | 52.03 | −3.78 | 0.00 | −299.60 | −93.55 | −299.60 | −93.55 |
| X Variable 3 | −167.52 | 52.02 | −3.22 | 0.00 | −270.52 | −64.52 | −270.52 | −64.52 |
| X Variable 4 | 106.64 | 43.94 | 2.43 | 0.02 | 19.63 | 193.65 | 19.63 | 193.65 |
| X Variable 5 | −19.76 | 39.39 | −0.50 | 0.62 | −97.76 | 58.25 | −97.76 | 58.25 |
| X Variable 6 | 2.44 | 34.45 | 0.07 | 0.94 | −65.77 | 70.66 | −65.77 | 70.66 |
| X Variable 7 | 42.85 | 29.04 | 1.48 | 0.14 | −14.66 | 100.35 | −14.66 | 100.35 |
| X Variable 8 | 172.66 | 58.40 | 2.96 | 0.00 | 57.02 | 288.30 | 57.02 | 288.30 |
| X Variable 9 | −269.36 | 54.24 | −4.97 | 0.00 | −376.77 | −161.96 | −376.77 | −161.96 |

To make the experimental results more evident, we calculated the fruit contour gradient along the X-axis projection using the tangent function formula and plotted it on a graph. The fruits were divided into four groups based on their rotation angles around the X-axis: 0, 5, 15, 30, 45, 60, 75, and 90 degrees. After performing linear fitting, we found that their $R^2$ coefficients were quite satisfactory. For example, the fitting coefficient for the first group (0 degrees) was 0.9992, and for the 5-degree group, it was 0.998. As depicted in Figure 11a, it becomes evident that the lychee fruits exhibit relatively minor angle variations, as evidenced by their slopes in the linear fitting, which are approximately −0.0076 and −0.0077. This suggests that differentiation based on both slope and intercept remains viable via the utilization of gradient distribution. Upon closer examination, as illustrated in Figure 11b–d, it becomes apparent that as the rotation angle of the fruit increases, there is a noticeable trend in the slopes obtained from linear fitting. Initially, these slopes increase and then subsequently decrease. This phenomenon occurs due to the projection of the fruit contour onto the X-axis, which corresponds to variations in rotation angles.

To conduct a comparative experiment, we projected the contours along the Y-axis with the morphological center as the midpoint. Similarly, we sampled nine points along the contour edge in sequential order. The correlation coefficient between the fruit contour rotation angle and the gradient value distribution was found to be 0.997, with a standard error of 0.411. As shown in Table 4, most of the standard errors were below eight units. The majority of the parameters had a *p* value less than 0.05, indicating that the model achieved significance or a confidence level of 95% at $\alpha = 0.05$. The research results demonstrate that whether projecting the lychee fruit contour along the X-axis or the Y-axis, a mathematical

correlation model can be obtained between the edge contour gradient and the fruit tilt angle. However, the standard error in the Y-axis projection was evidently more favorable.

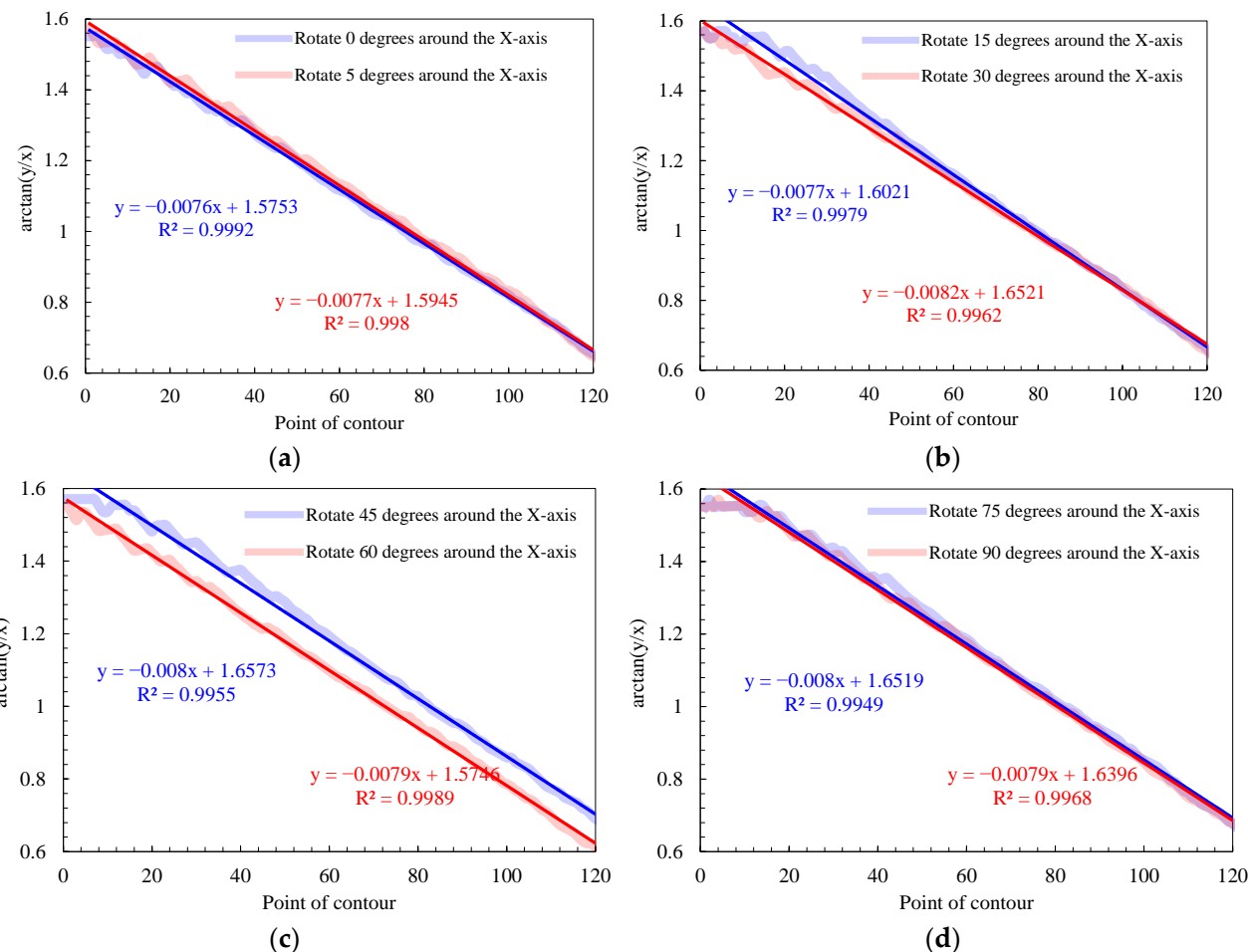

**Figure 11.** Distribution of contour gradient along the X-axis from different angles ((**a**) represents fruit rotation from 0 to 5 degrees, (**b**) represents fruit rotation from 13 to 30 degrees, (**c**) represents fruit rotation from 45 to 60 degrees, and (**d**) represents fruit rotation from 75 to 90 degrees).

**Table 4.** Regression parameters of Y-axis.

|  | Coefficients | Standard Error | t Stat | *p*-Value | Lower 95% | Upper 95% | Upper Limit 95.0% | Lower Limit 95.0% |
|---|---|---|---|---|---|---|---|---|
| Intercept | 70.00 | 16.09 | 4.35 | 0.00 | 37.72 | 102.29 | 37.72 | 102.29 |
| X Variable 1 | 71.12 | 7.66 | 9.29 | 0.00 | 55.76 | 86.49 | 55.76 | 86.49 |
| X Variable 2 | 4.18 | 2.80 | 1.49 | 0.14 | −1.45 | 9.80 | −1.45 | 9.80 |
| X Variable 3 | −23.42 | 7.38 | −3.17 | 0.00 | −38.23 | −8.61 | −38.23 | −8.61 |
| X Variable 4 | 9.87 | 3.53 | 2.79 | 0.01 | 2.78 | 16.96 | 2.78 | 16.96 |
| X Variable 5 | −20.58 | 6.64 | −3.10 | 0.00 | −33.90 | −7.26 | −33.90 | −7.26 |
| X Variable 6 | −10.30 | 3.20 | −3.22 | 0.00 | −16.73 | −3.88 | −16.73 | −3.88 |
| X Variable 7 | 0.11 | 4.73 | 0.02 | 0.98 | −9.38 | 9.61 | −9.38 | 9.61 |
| X Variable 8 | 1.84 | 2.49 | 0.74 | 0.46 | −3.16 | 6.85 | −3.16 | 6.85 |
| X Variable 9 | −26.27 | 5.45 | −4.82 | 0.00 | −37.20 | −15.34 | −37.20 | −15.34 |

Similarly, we plotted the gradient of the fruit contour projected along the Y-axis. Unlike the analysis of the X-axis projection gradient, here we used polynomial fitting, and we found that the maximum $R^2$ coefficient reached 0.9748. As shown in Figure 12a, due to the relatively small range of lychee fruit tilt angles, their $R^2$ values in polynomial fitting were 0.9327 and 0.9389, respectively. Upon observation, in Figure 12b–d, it was found that the gradient distribution between 15 degrees and 60 degrees of fruit rotation angle exhibited a relatively ideal level of differentiation, while the gradient distribution showed little difference between 0 degrees and 5 degrees or between 75 degrees and 90 degrees.

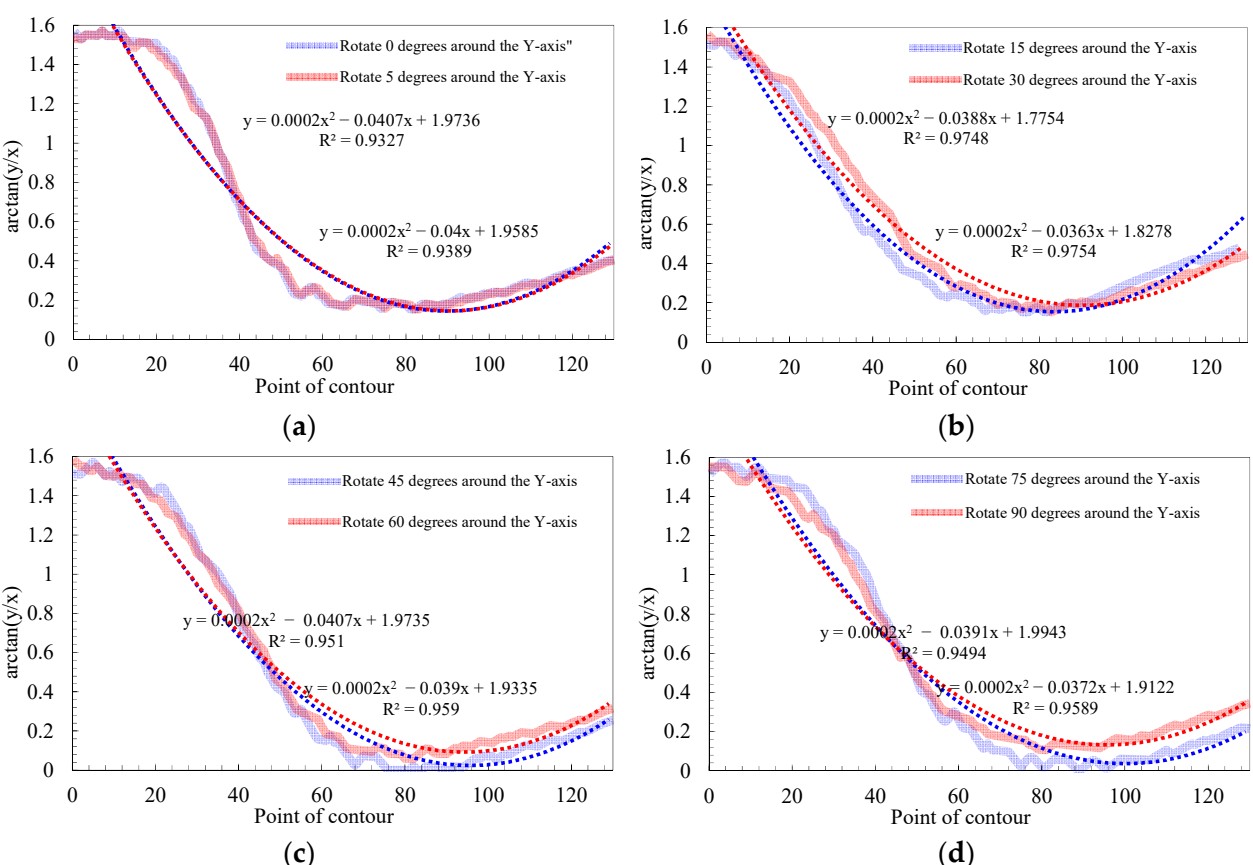

**Figure 12.** Distribution of contour gradient along the Y-axis from different angles.

### 3.4. Euclidean Distance Positioning Accuracy

Euclidean distance refers to the distance between two points in Euclidean space. In two or three-dimensional space, the Euclidean distance between two points with coordinates $(x_1, y_1)$ and $(x_2, y_2)$ can be calculated. We extracted 222 occluded lychee cluster images and 224 non-occluded lychee cluster images for pixel accuracy experiments. To provide a more intuitive representation of the distribution of distance errors, we calculated their Euclidean distance between predicted and ground truth picking points. We created a histogram, as depicted in Figures 13 and 14. Our analysis indicates that for type A picking point with occlusion, most of the distance errors associated with predicted picking points were below 100 pixels, while for type A without occlusion, most of these distance errors were less than 80 pixels. The accumulation curve is represented as the red line. In addition, for type B picking point, most of the distance errors associated with predicted picking points were below 100 pixels, while for type A without occlusion, most of these distance errors were less than 90 pixels. It is worth noting that the size of the images we collected is $1440 \times 1080$, which demonstrates the effectiveness of using the distribution of passed lychee masks for locating picking points.

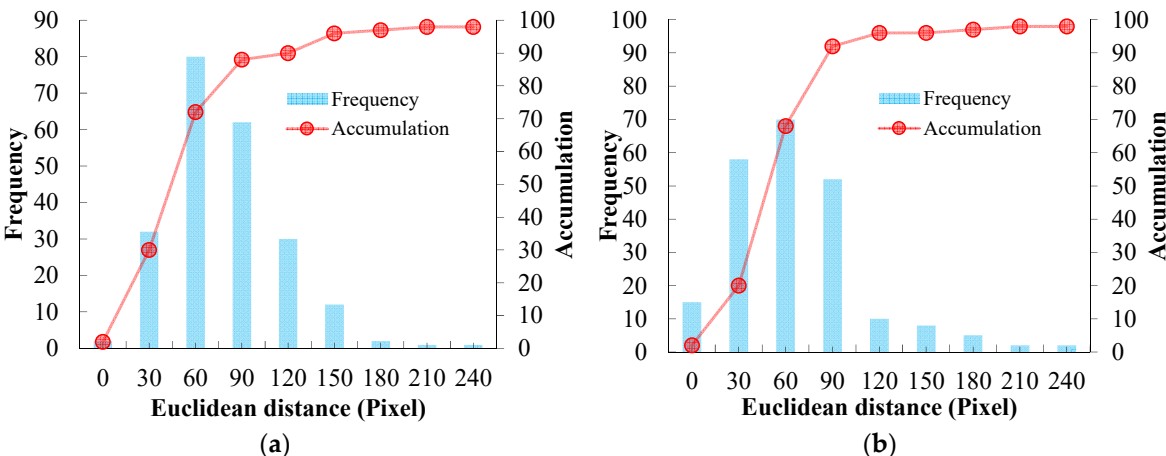

**Figure 13.** Histogram displaying the distribution of the Euclidean distances between the predicted picking points and ground truth values: (**a**) type A with leaf occlusion and (**b**) type A without leaf occlusion.

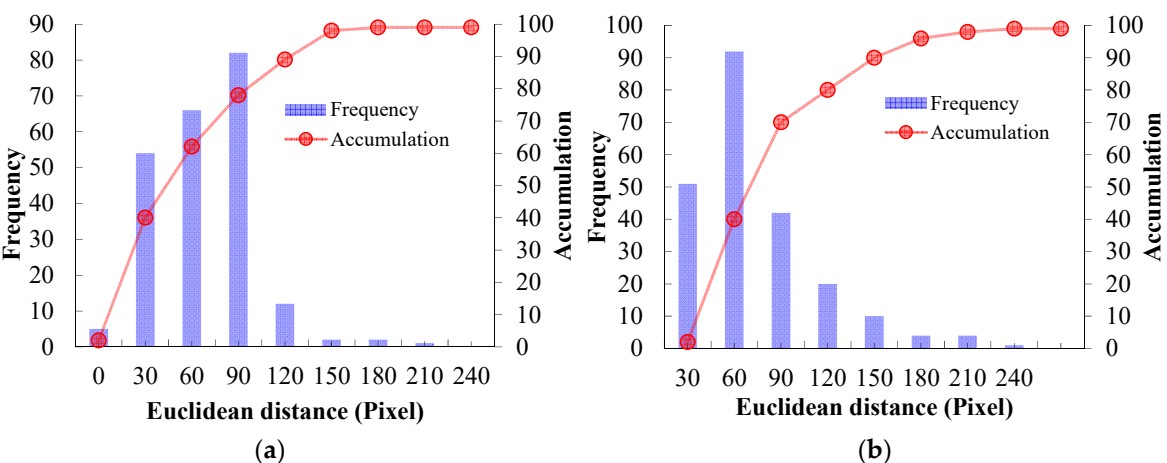

**Figure 14.** Histogram displaying the distribution of the Euclidean distances between the predicted picking points and ground truth values: (**a**) type B with leaf occlusion and (**b**) type B without leaf occlusion.

### 3.5. Position Accuracy Evaluation of LP³Net

Obviously, the number of lychees will affect the positioning success rate. According to the empirical value, this paper divides the fault-tolerance accuracy of lychee picking targets into single, cluster positioning. As shown in Figure 15, we compared FCIS, LP³Net, Mask RCNN, and Center Mask for mAP and speed on random datasets and evaluated the detection via single and cluster litchi, respectively. As can be seen, LP³Net achieved the best mAP, followed by Center Mask with 76.43% mAP. The detection head of LP³Net can simultaneously predict the category score, bounding box regression parameters, and mask coefficient. In terms of the FPS comparison effect, LP³Net still reached the highest 30 fps. Although the mAP of LP³Net is the highest when the IoU is 50, when the IoU value is 65 or 80, LP³Net has a significant effect on improving the accuracy. In summary, the following will focus on comparing the detection effects of different versions of LP³Net so that we can select the fastest and best detector for target tracking of a cluster of litchi.

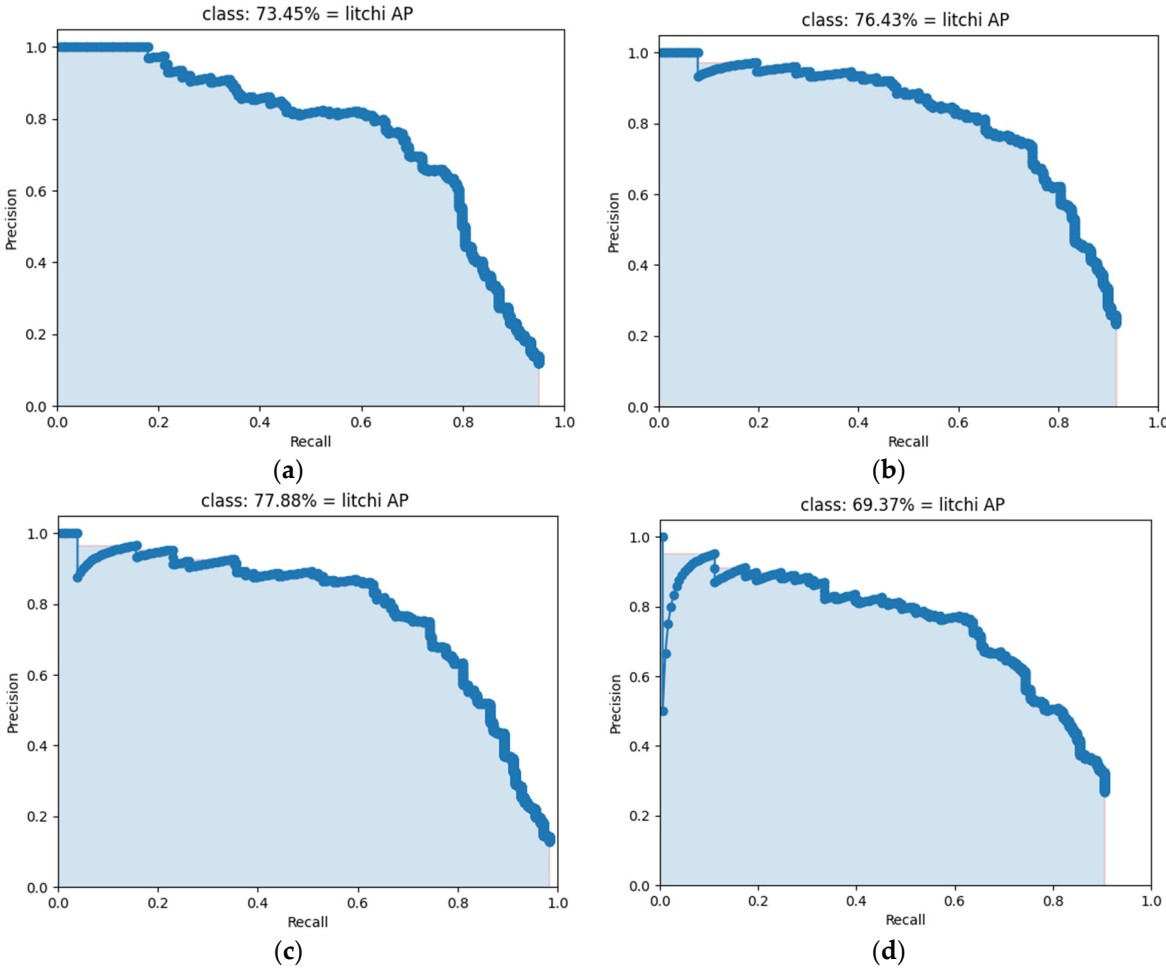

**Figure 15.** Precision–recall curve of 4 kinds of methods with IoU is 50: (**a**) FCIS, (**b**) Mask-RCNN, (**c**) LP$^3$Net, and (**d**) Center Mask.

Next, we will perform performance statistics based on the number of lychee fruits in each group via LP$^3$Net. Here, 200 tracking targets were evaluated, respectively, with PP position, AP position, and MISS as statistical objects, as shown in Table 5, in the single-fruit test, the number of target points with predicted points falling within the radius R$_1$ is 118, and the success rate of target point positioning is 95%. This shows that the lychee single fruit picking point has obvious characteristics, and the target location mechanism can easily obtain high-precision information. When the number of bunched fruit is greater than 2 and less than 5, the success rate is only 72.5%, which shows that there are not many lychee bunches, but it is still difficult to pick and locate because the small number of single lychee fruit represents the normal direction obtained via the mask in small quantities. When the number of lychee bunches is greater than 5, the target localization can achieve a success rate of 81%, which shows the effectiveness of the multi-target tracking and localization method proposed in this paper.

**Table 5.** Target numbers with different numbers of lychee brunch. (SR: success rate; MR: miss rate; Bet.2–5: the number of lychee is between 2 to 5).

| Type | PP | AP | Miss | SR | MR |
|---|---|---|---|---|---|
| Single | 118 | 72 | 10 | 95% | 5% |
| Bet.2–5 | 85 | 60 | 55 | 72.50% | 27.50% |
| Above 5 | 90 | 72 | 38 | 81% | 19% |

When the diameter of the lychee stem is occluded, the target fault-tolerance method based on the Mask NV proposed in this paper can be combined with the contour features of the lychee to predict the picking point. Here, the target positioning effect of the two types of picking scenes A and B is shown in Figure 16. This includes the single and multi-target positioning of single and bunched fruit. Figure 16a,b are the positioning effects of single fruit and normality distribution, respectively. Figure 16c,d are the positioning effects of the right and the left distribution, respectively.

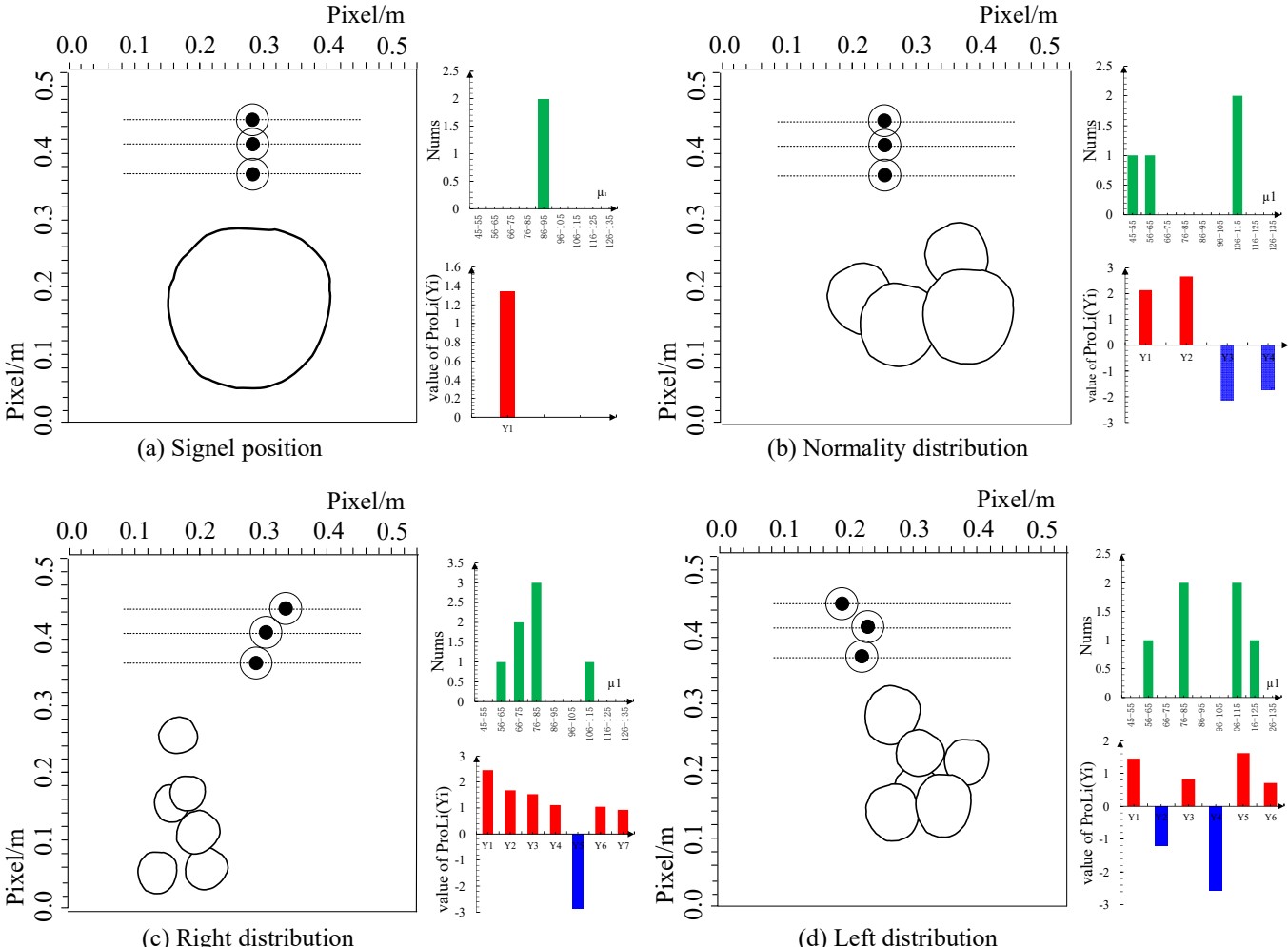

**Figure 16.** Single brunch localization (The green histogram represents the distribution of the quantity for each object at a certain inclined angle, while the red and blue histograms represent the projection lengths on the coordinate axes).

Figure 17 contains multiple picking targets, which are numbered according to the picking sequence with the upper left corner of the image as the origin. Target 1 in the figure is single fruit picking, and targets 2 and 3 in the figure are cluster fruit picking. It can be seen from the figure that, although different, the algorithm in this paper can still locate the diameter of the picking rod within a certain range.

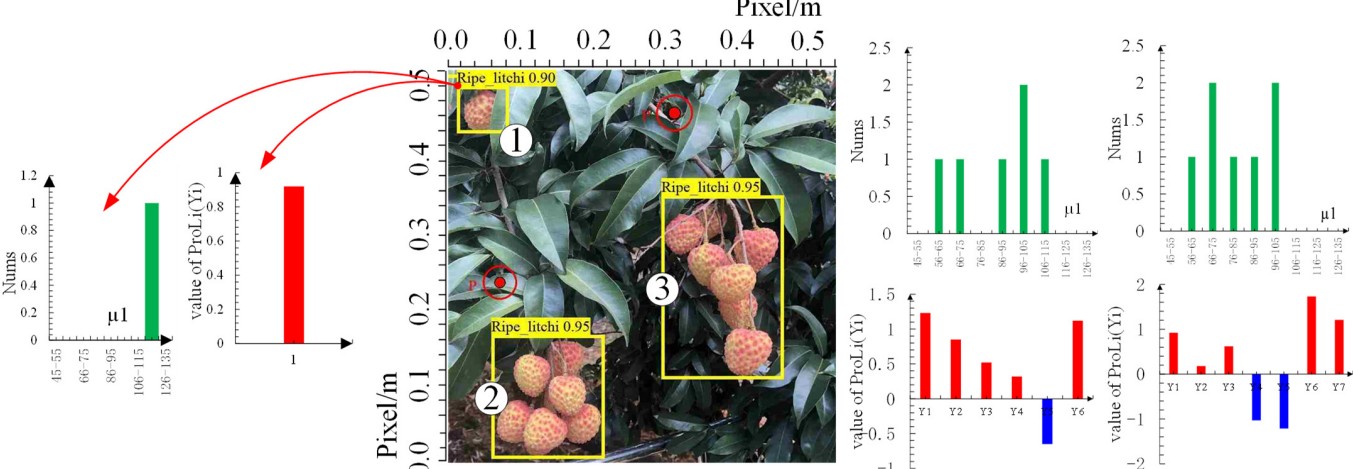

**Figure 17.** Prediction of obstructing lychee picking points (1, 2, and 3 represent the identification numbers of the lychee fruit clusters).

Figure 18 shows the positioning effect of the picking points in grayscale images under random distribution. It can be seen from the figure that the algorithm proposed in this paper can perform single lychee segmentation well and can predict the distribution of brunch picking points accurately.

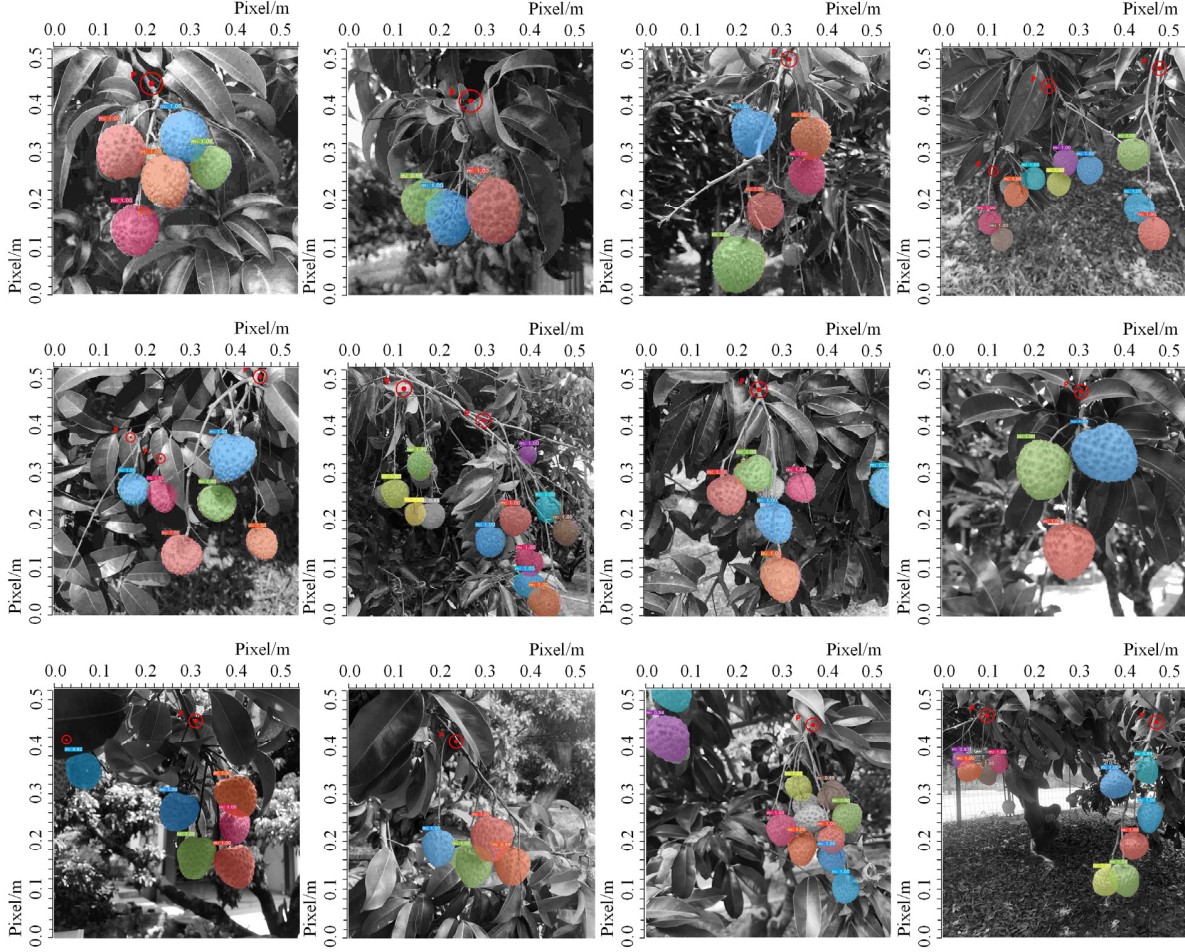

**Figure 18.** Instance segmentation and picking point location under different distributions.

### 3.6. Accuracy Evaluation with RGB-D Information

The lychee 3D locations were easily obtained by matching its corresponding pixel-in-depth images with RGB mask position. After matching the information of the camera, the target picking will use RGB-D data association to mark and count the depth information of lychee and then obtain $p_z$ [71,72]. In addition, the $p_x$ and $p_y$ coordinates of picking point $P$ are a set of interval values, but the $p_z$ coordinate is a fixed value derived from an unbiased estimator of the target point interval. In order to evaluate 3D positioning accuracy, it was tested at different times [17,73,74]. Assuming that positioning radius $R_2$ is taken as the benchmark here, if the circle drawn can include the lychee string rod diameter, then it is a successful try. In Figure 19 below, the yellow histogram is a successful location rate without LP$^3$Net. On the contrary, the blue part is the successful location rate with our mechanism. In the case of 50 attempts, accuracy is compared here. Initially, the success rate of the type A position is 82% without the application of LP$^3$Net, whereas it increases to 92% when the LP$^3$Net and fault-tolerance are implemented. In cases where occlusion is present, the success rate for locating the target of class A can reach 70%. As the number of attempts increases, the fault-tolerance mechanism has demonstrated a relatively high success rate in localizing targets of both class A and class B during the picking scenarios.

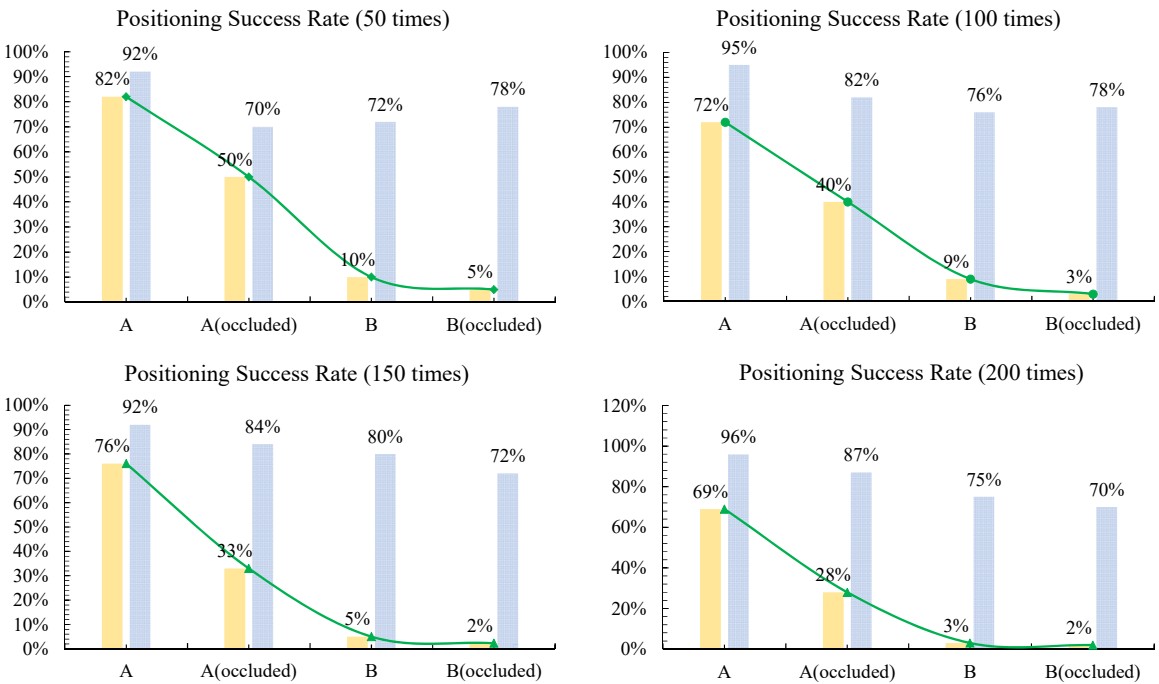

**Figure 19.** Success position rate with RGB-D information (A represents lychee clusters picked from unobstructed and non-tilted locations, A (occluded) represents lychee clusters picked from obstructed and non-tilted locations, B represents lychee clusters picked from unobstructed and tilted locations, and B (occluded) represents lychee clusters picked from obstructed and tilted locations).

## 4. Conclusions

The aim of this paper is to combine the phenotypic characteristics of lychee fruit clusters with artificial intelligence algorithms to propose a method for predicting the position of obscured pedicels. The instance segmentation of artificial intelligence algorithms often fails to separate the pedicel and fruit bunch obstructed by leaves, which greatly hinders the development of automated lychee harvesting technology. Therefore, the proposed method in this paper provides a fundamental breakthrough and excellent technical support for image processing of clustered fruits. In summary, the key findings of this paper can be outlined as follows:

(1) This paper introduces LP³Net, an end-to-end prediction network designed to locate pedicels in clustered fruits. LP³Net offers several advantages, including the ability to delineate the contours of partially obscured fruits, generate high-quality instance masks, and provide stable real-time localization, all without relying on repooling.

(2) This research identifies a limitation in instance segmentation models when comparing lychee fruit features to fixed patches. To enhance overall model performance, this paper proposes the incorporation of patch features using XProtoNet as part of the model prediction.

(3) This paper delves into the analysis of gradient direction distribution within lychee fruit contours and presents a regression analysis of the gradient histogram relative to the frontal view's picking point position. The findings reveal a consistent mathematical model describing the relationship between fruit edge contour gradients and fruit inclination angles. Notably, projections of gradient vectors along the Y-axis yield more accurate results in terms of standard error. The gradient distribution effectively discriminates between fruit rotation angles ranging from 15 to 60 degrees, while exhibiting less variability between 0 and 5 degrees or 75 and 90 degrees.

**Author Contributions:** Methodology, J.L.; Validation, J.W.; Data curation, Y.L. (Yangfan Luo); Writing—original draft, Y.L. (Yuanhong Li); Project administration, Y.L. (Yubin Lan). All authors have read and agreed to the published version of the manuscript.

**Funding:** This work was financially supported by the Laboratory of Lingnan Modern Agriculture Project (Grant No. NT2021009), the National Natural Science Foundation of China (Grant No. 32301708), Guangdong Basic and Applied Basic Research Foundation (Grant No. 2021A1515110554), Key-Area Research and Development Program of Guangdong Province (No. 2019B020214003) and the 111 Project (D18019), China Postdoctoral Science Foundation (Grant No. 2022M721201), China Agriculture Research System (CARS-15-23), and The Open Competition Program of the Top Ten Critical Priorities of Agricultural Science and Technology Innovation for the 14th Five-Year Plan of Guangdong Province (No. 2022SDZG03).

**Data Availability Statement:** Not applicable.

**Conflicts of Interest:** The authors declare no conflict of interest.

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
