# Peer review of "Prototype Network for Predicting Occluded Picking Position Based on Lychee Phenotypic Features"

_agronomy, doi:10.3390/agronomy13092435_

Round 1

Reviewer 1 Report

The manuscript, "Prototype Network for Predicting Occluded Picking Position Based on Lychee Phenotypic Features," tackles an important issue in automated fruit harvesting with a focus on lychee fruits. While the authors have shown an effort to address the topic through LP3Net, several significant shortcomings should be addressed for a manuscript of this scope and ambition.

  1. Technical Specificity: A more detailed exposition of technical components is needed. For example, the manuscript can benefit from diving into the underlying mathematical models, akin to the approach used in "Output-Feedback Robust Tracking Control of Uncertain Systems via Adaptive Learning."

  2. Comparative Discussion: The work seems to exist in a vacuum, with scant reference to other state-of-the-art techniques. Comparative analysis with advanced methodologies like "EGNN: Graph structure learning based on evolutionary computation helps more in graph neural networks" could provide a richer context for the paper's contributions.

  3. Clarity of Contribution: The manuscript's contributions relative to existing work remain unclear. Given the current state of artificial intelligence in agriculture, elucidated in papers like "Arc Fault Detection Using Artificial Intelligence: Challenges and Benefits," understanding how this work differentiates itself becomes crucial.

  4. Complexity and Nonlinearity: The control mechanisms implemented appear to be rather linear. A more comprehensive approach could include aspects of nonlinearity, much like what's discussed in "Center-based Transfer Feature Learning With Classifier Adaptation for surface defect recognition."

  5. Sensor Interactions: As the manuscript touches upon artificial intelligence algorithms for image processing, some reference or adaptation of work like "self-powered difunctional sensors based on sliding contact-electrification and tribovoltaic effects for pneumatic monitoring and controlling" could add a layer of depth, considering their applicability in real-world conditions.

  6. Instance Segmentation: It would be beneficial to delve deeper into the mechanisms of instance segmentation, particularly in challenging conditions. Here, a parallel could be drawn to "Heterogeneous Network Representation Learning Approach for Ethereum Identity Identification," which deals with complex, non-linear data distributions.

  7. Real-world Applicability: The manuscript could be strengthened by discussions or case studies involving the technology's actual field deployment, offering more concrete evidence of its applicability and scalability.

  8. Mathematical Consistency: While the manuscript discusses the mathematical relationship between contour edge gradients and their inclination angles, it would be more persuasive if these calculations are deeply anchored in existing mathematical principles or theories.

In summary, while the work is promising and aims to tackle an important issue, there are substantial gaps both in the detailing of technical methodologies and the comparative discussion with existing works. Therefore, major revisions are recommended.

Reviewer 2 Report

The paper addresses the challenge of automated harvesting of clustered fruits, particularly focusing on fruits that are obscured by leaves and have stem diameters that lack discernible texture patterns. I have following observations.

1. Choosing appropriate lychee phenotypic features that encapsulate the necessary information for accurate occluded picking position prediction is crucial. How do authors deal with such issue?

2. The paper lacks mathematical support to the proposed method. Relevant mathematical equations amy be used.

3. The paper suggests calculating fruit normal vectors by employing edge computation and analyzing the distribution of gradient directions in the fruit contour. The details of these calculations are missing in the paper.

4. The authors introduce a fully convolutional, feature prototype-based one-stage instance segmentation network designed for occluded lychee clusters. Limited number of performance metrics are used. Additional relevant metrics may be used.

Round 2

Reviewer 1 Report

The authors have improved a lot based on the comments. But the reference seems not to be updated when you consider the corresponding related works. Please check and update the manuscript before resubmission. 

Author Response

Thank you very much for your suggestions. I have made the necessary revisions to the paper as per your instructions and updated the references.

[19] Zhao, J.; Lv, Y. Output-Feedback Robust Tracking Control of Uncertain Systems via Adaptive Learning. Int. J. Control Autom. Syst. 2023, 21, 1108–1118, doi:10.1007/s12555-021-0882-6.

[20] Liu, Z.; Yang, D.; Wang, Y.; Lu, M.; Li, R. EGNN: Graph Structure Learning Based on Evolutionary Computation Helps More in Graph Neural Networks. Appl. Soft Comput. 2023, 135, 110040, doi:10.1016/j.asoc.2023.110040.

[21] Wang, Y.; Liu, Z.; Xu, J.; Yan, W. Heterogeneous Network Representation Learning Approach for Ethereum Identity Identification. IEEE Trans. Comput. Soc. Syst. 2023, 10, 890–899, doi:10.1109/TCSS.2022.3164719.

[22] Tian, C.; Xu, Z.; Wang, L.; Liu, Y.; Tian, C.; Xu, Z.; Wang, L.; Liu, Y. Arc Fault Detection Using Artificial Intelligence: Challenges and Benefits. Math. Biosci. Eng. 2023, 20, 12404–12432, doi:10.3934/mbe.2023552.

[23] Shi, Y.; Li, L.; Yang, J.; Wang, Y.; Hao, S. Center-Based Transfer Feature Learning With Classifier Adaptation for Surface Defect Recognition. Mech. Syst. Signal Process. 2023, 188, 110001, doi:10.1016/j.ymssp.2022.110001.

[24] Shi, Y.; Li, H.; Fu, X.; Luan, R.; Wang, Y.; Wang, N.; Sun, Z.; Niu, Y.; Wang, C.; Zhang, C.; et al. Self-Powered Difunctional Sensors Based on Sliding Contact-Electrification and Tribovoltaic Effects for Pneumatic Monit

Reviewer 2 Report

The required changes are made 

Author Response

Thanks.